# COPD Exacerbation-Related Pathogens and Previous COPD Treatment

**DOI:** 10.3390/jcm12010111

**Published:** 2022-12-23

**Authors:** Yun Su Sim, Jin Hwa Lee, Eung Gu Lee, Joon Young Choi, Chang-Hoon Lee, Tai Joon An, Yeonhee Park, Young Soon Yoon, Joo Hun Park, Kwang Ha Yoo

**Affiliations:** 1Division of Pulmonary, Allergy, and Critical Care Medicine, Department of Internal Medicine, Hallym University Kangnam Sacred Heart Hospital, Seoul 07441, Republic of Korea; 2Division of Pulmonary and Critical Care Medicine, Department of Medicine, Ewha Womans University College of Medicine, Seoul 07804, Republic of Korea; 3Bucheon St. Mary’s Hospital, College of Medicine, The Catholic University of Korea, Bucheon 14647, Republic of Korea; 4Division of Pulmonary and Critical Care Medicine, Department of Internal Medicine, Incheon St. Mary’s Hospital, College of Medicine, The Catholic University of Korea, Seoul 21431, Republic of Korea; 5Division of Pulmonary and Critical Care Medicine, Department of Internal Medicine, Seoul National University Hospital, Seoul 03080, Republic of Korea; 6Division of Pulmonary and Critical Care Medicine, Department of Internal Medicine, Yeouido St. Mary’s Hospital, College of Medicine, The Catholic University of Korea, Seoul 07345, Republic of Korea; 7Division of Pulmonary and Critical Care Medicine, Department of Internal Medicine, Daejeon St. Mary’s Hospital, College of Medicine, The Catholic University of Korea, Seoul 34943, Republic of Korea; 8Division of Pulmonary and Critical Care Medicine, Department of Internal Medicine, Dongguk University Ilsan Hospital, Goyang 10326, Republic of Korea; 9Department of Pulmonary and Critical Care Medicine, Ajou University School of Medicine, Suwon 16499, Republic of Korea; 10Department of Internal Medicine, Konkuk University School of Medicine, Seoul 05030, Republic of Korea

**Keywords:** respiratory pathogen, chronic obstructive pulmonary disease, inhaler

## Abstract

We evaluated whether the pathogens identified during acute exacerbation of chronic obstructive pulmonary disease (AE-COPD) are associated with the COPD medications used in the 6 months before AE-COPD. We collected the medical records of patients diagnosed with AE-COPD at 28 hospitals between January 2008 and December 2019 and retrospectively analyzed them. Microorganisms identified at the time of AE-COPD were analyzed according to the use of inhaled corticosteroid (ICS) and systemic steroid after adjusting for COPD severity. We evaluated 1177 patients with AE-COPD and available medication history. The mean age of the patients was 73.9 ± 9.2 years, and 83% were males. The most frequently identified bacteria during AE-COPD were *Pseudomonas aeruginosa* (10%), followed by *Mycoplasma pneumoniae* (9.4%), and *Streptococcus pneumoniae* (5.1%), whereas the most commonly identified viruses were rhinovirus (11%) and influenza A (11%). During AE-COPD, bacteria were more frequently identified in the ICS than non-ICS group (*p* = 0.009), and in the systemic steroid than non-systemic steroid group (*p* < 0.001). In patients who used systemic steroids before AE-COPD, the risk of detecting *Pseudomonas aeruginosa* was significantly higher during AE-COPD (OR 1.619, CI 1.007–2.603, *p* = 0.047), but ICS use did not increase the risk of Pseudomonas detection. The risk of respiratory syncytial virus (RSV) detection was low when ICS was used (OR 0.492, CI 0.244–0.988, *p* = 0.045). COPD patients who used ICS had a lower rate of RSV infection and similar rate of *P. aeruginosa* infection during AE-COPD compared to patients who did not use ICS. However, COPD patients who used systemic steroids within 6 months before AE-COPD had an increased risk of *P. aeruginosa* infection. Therefore, anti-pseudomonal antibiotics should be considered in patients with AE-COPD who have used systemic steroids.

## 1. Introduction

Chronic obstructive pulmonary disease (COPD) is increasing worldwide with the aging of the population [1,2,3]. The treatment goals of stable COPD are to reduce the current symptoms and risk of disease exacerbation [1]. The treatment of stable COPD is based on the inhalation of long-acting bronchodilators alone or in combination with long-acting beta 2-agonist (LABA), and long-acting muscarinic antagonist (LAMA) [1]. Inhaled corticosteroid (ICS) with anti-inflammatory action is used in combination with other inhaled bronchodilators to improve lung function and health status and reduce exacerbations in patients with moderate to very severe COPD [1].

Acute exacerbation of COPD (AE-COPD) is closely related to the quality of life, prognosis, and mortality of COPD patients [1]. AE-COPD is often caused by respiratory infections [4,5]. Prospective cohort studies identified an infectious aetiology in 88% of cases of AE-COPD [6]. Since the infectious pathogen of COPD patients can be affected by the duration of the disease, the patient’s comorbidties, and the previous exacerbation history, various factors of AE-COPD patients should be considered when selecting antibiotics for appropriate treatment [1,7,8]. Previous medications used in COPD patients could also be important factors affecting respiratory pathogens detected in AE-COPD.

Previous studies [9,10,11,12,13,14,15,16] have shown that ICS, which has anti-inflammatory and immunosuppressive effects increase the risks of pneumonia, tuberculosis, and non-tuberculosis mycobacterial lung disease. A case–control study of patients with AE-COPD showed that ICS dose, but not use, increased the risk of *Pseudomonas aeruginosa* infection in severe COPD [17]. However, few studies have evaluated the effects of various inhaled agents for COPD on respiratory bacterial or viral infections. Therefore, we investigated whether the pathogens identified during AE-COPD are associated with the COPD medications used for 6 months before AE-COPD.

## 2. Materials and Methods

### 2.1. Study Design

This study is retrospective multicenter cohort study. This study was performed at 28 hospitals between January 2015 and December 2018 in the Republic of Korea. We collected the medical records of patients diagnosed with AE-COPD and retrospectively analyzed them. The criteria for the subject of this study: (1) aged > 40 years, (2) history of COPD diagnosed with post-bronchodilator forced expiratory volume in 1 s [FEV1]/forced vital capacity < 0.7, (3) diagnosis of moderate-to-severe AE-COPD, and (4) all conventional tests for detect the causative pathogen in AE-COPD.

Moderate-to-severe AE-COPD was defined as on the basis of the need for additional medication or hospitalization due to worsening clinical symptoms such as cough, dyspnea, and sputum based on the definition of the Global Initiative for Obstructive Lung Disease (GOLD) guidelines [18].

This study was approved by the Institutional Review Board of Hallym University Kangnam Sacred Heart Hospital (HKS 2019-12-016-002). Patient information was anonymized and de-identified before analysis; therefore, the requirement for informed consent was waived. This study was conducted in accordance with the 2013 revision of the Declaration of Helsinki.

### 2.2. Variables

We assessed the demographic and clinicopathological information of patients including age, sex, body mass index, comorbidities, lung function test results and medication use before AE-COPD.

The oral medications included xanthine derivatives, leukotriene receptor antagonists, and systemic corticosteroids. The inhaled treatments included intermittent short-acting beta 2-agonists (SABA), LABA, LAMA, ICS, and combinations of those for 6 months before AE-COPD.

The microbiological examination included Gram-stain and culture of sputum or endotracheal aspirates, sputum polymerase chain reaction (PCR) for *Bordetella pertussis*, *Chlamydophila pneumoniae*, *Mycoplasma pneumoniae*, and *Legionella pneumophila* and viruses, serum antibody tests for *C. pneumoniae* and *M. pneumoniae*, and nasal swab tests for influenza A and B virus antigens. Pathogen classification was identified according to microbiological test results, single or multiple virus or bacterial infection, no pathogen detected.

The severity of COPD was classified by using GOLD grade and the pulmonary function test in the patients when GOLD grades were not investigated. GOLD A or FEV_1_ 80% or more is mild group, GOLD B or FEV_1_ less than 80% to 50% or more is moderate group, GOLD C or FEV_1_ less than 50% to 30% or more is severe group, GOLD D or FEV_1_ less than 30% is very severe group.

### 2.3. Statistical Analysis

In this study, cases where bacterial and viral tests were not performed were treated as missing value. Frequencies are expressed as numbers (%) and descriptive data are expressed as median value and interquartile range. The Chi-square test or Fisher’s exact test were used for categorical variables, and continuous variables were compared using Kruskal-wallis test. Statistical significance was set at *p* < 0.05. The effect of inhaled corticosteroid or systemic steroid administration on bacterial or viral detection during exacerbation of chronic obstructive pulmonary disease was evaluated through logistic regression analysis with adjusted for the severity of COPD.

## 3. Results

We evaluated 1177 patients with AE-COPD and available medication histories. *C. pneumonia* and *M. pneumonia* were identified in serum via immunoglobulin M or analysis in respiratory specimens via PCR. The tests were performed in 739 cases, and cases that were not tested were treated as missing data. Influenza A and B were identified in respiratory specimen using PCR or immunofluorescence assay. The above tests were performed in 1131 cases were tested, and cases that were not tested were treated as missing data. Table 1 presents baseline demographic and clinical characteristics of the patients with AE-COPD according to the type of inhaler used. The mean age of patients with AE-COPD was 73.9 ± 9.2 years, and 83% were males. The mean durations of disease and treatment were longest in patient using the ICS/LABA/LAMA inhalers. The proportion of never smokers was highest in the ICS group and the lowest in the ICS/LABA/LAMA group. Lung function was worst in the ICS/LABA/LAMA group and best in ICS group. The use of systemic steroids was the most common in the ICS group.

Figure 1 shows the bacterial and viral species identified in patients with AE-COPD according to the type of inhaler used. Bacteria were identified in 32% of patients with AE-COPD. In 3.2% of patients, multiple bacteria were identified. The most commonly identified bacteria were *Pseudomonas aeruginosa* (10%), *Mycoplasma pneumoniae* (9.4%), *Klebsiella pneumoniae* (4.4%), and *Streptococcus pneumoniae* (5.1%). *P. aeruginosa* had the highest detection rates in the ICS/LABA/LAMA, LABA/LAMA, LABA, and SABA groups, whereas *M. pneumoniae* had the highest detection rates in the ICS/LABA, LAMA, ICS, and no inhaler use groups. Viruses were detected in 33% of patients with AE-COPD. In patients hospitalized for AE-COPD, the most commonly detected viruses were rhinovirus (11%) and influenza virus A (11%), followed by respiratory syncytial virus (RSV) (4.3%).

Figure 2 and Figure 3 present the bacterial and viral detection rates for patients with and without ICS and systemic steroid use, respectively. During AE-COPD, bacteria were more frequently identified in the ICS than non-ICS group (*p* = 0.009), and in the systemic steroid than non-systemic steroid group (*p* < 0.001). Because the treatment for COPD depends on disease severity, logistic regression analysis was performed to determine the effects of ICS and systemic steroid use on the detection rate of each pathogen after adjusting for COPD severity (Table 2). The *P. aeruginosa* detection rate was not increased with ICS use (odds ratio [OR] = 1.238, 95% confidence interval [CI] = 0.796–1.927, *p* = 0.343), but was increased with systemic steroid use (OR = 1.619, 95% CI = 1.007–2.603, *p* = 0.047). The rate of RSV detection was decreased with ICS use (OR = 0.492, 95% CI = 0.244–0.988, *p* = 0.045).

## 4. Discussion

In the present study, the most commonly identified bacteria during AE-COPD were *P. aeruginosa*. Previous microbiological analyses of AE-COPD patients detected *P. aeruginosa* most frequently [6,19]. In this study, *P. aeruginosa* was detected in 13% of patients receiving ICS/LABA/LAMA triple therapy, whereas it was detected in only 6.9% of patients with AE-COPD who were not using inhalers. GOLD grade D patients accounted for 30% and 1.8% of the ICS/LABA/LAMA and no inhaler groups, respectively. The number of acute exacerbations and *P. aeruginosa* identification rate were increased in the ICS/LABA/LAMA group, which included a greater proportion of GOLD grade D patients.

COPD treatment depends on disease severity. Microbiological analysis may provide biased results depending on the severity of previous exacerbations. Therefore, in the present study, we analysed the microbiological results of patients with previous ICS and systemic steroid use with adjustment of COPD severity.

In our study, ICS use did not increase the *P. aeruginosa* detection rate. In a study of 60 patients with stable moderate COPD [20], long-term ICS use influenced the airway bacterial load and low eosinophil counts were associated with increased airway bacterial load. In a 4-year study of 380 COPD patients [17] and an epidemiological cohort study of Danish COPD patients [21], the risk of *P. aeruginosa* infection varied with ICS dose, but not with its use. Some studies have found that the risk of pneumonia, tuberculosis, and mycobacterial disease increases after long-term ICS use [9,10,11,12,13,14,15,16]. Conversely, a prospective randomized study of 237 COPD patients [22] found that the use of high-dose ICS did not significantly increase the incidence of pneumonia. An experimental study of human lung tissue [23] showed that budesonide inhibits intracellular infection with non-typeable *Haemophilus influenzae* by suppressing p38 MAPK. Given the contradictory findings of previous studies, it is unclear whether ICS increases the risk of bacterial infections in COPD patients. Although additional studies are required on the relationships of pathogens with the ICS dose and eosinophil count in AE-COPD patients, ICS use may not have a significant association with virulent or drug-resistant bacteria during AE-COPD.

In our study, the *P. aeruginosa* detection risk was significantly increased in patients using systemic steroid after adjusting for COPD severity. These findings are consistent with previous studies [24,25]. A study [24] of 188 patients with AE-COPD showed that the *P. aeruginosa* detection risk increased according to the amounts of systemic steroids used in patients hospitalized with AE-COPD. Another study [25] of hospitalized AE-COPD patients found that systemic steroid use was an independent risk factor for *K. pneumoniae* and *P. aeruginosa* infection. Long-term use with steroids weakens adaptive immune response by down-regulation MHC class II and costimulatory molecules [26]. Systemic steroid treatment was also reported to be associated with poor clearance of causative microbiologic pathogen of AE-COPD following antibiotic treatment [27].

Previous systemic steroid use indicates a history of COPD exacerbation. AE-COPD is a risk factor for future AE-COPD [28], Additionally, the high rate of detection of bacteria and *P. aeruginosa* in patients with systemic steroid use might increase the likelihood of future exacerbations [29]. Long-term use of corticosteroids decreased the adaptive immune response by down-modulating the major histocompatibility complex (MHC) class II and costimulatory molecules [26].

AE-COPD caused by *P. aeruginosa* has a high mortality rate. A single institution study of AE-COPD patients showed that systemic steroids were not prescribed in accordance with evidence-based recommendations [30]. Because systemic steroids increase the *P. aeruginosa* detection rate, the dose and duration of use of systemic steroids during AE-COPD should be carefully considered; furthermore, the use of ICS as an adjunctive treatment should be considered.

In our study, *M. pneumoniae* was most commonly detected during AE-COPD in patients who used ICS or ICS/LABA. However, some previous studies did not detect any AE-COPD patients with *M. pneumonia* [31,32,33]. In other studies [34,35], *M. pneumonia* was detected in 4.7–5% of AE-COPD patients, with no significant difference in the detection rate according to clinical characteristics or COPD severity [35]. Although it is important to administer antibiotics that can address *P. aeruginosa* infection during AE-COPD, the high detection rate of *M. pneumoniae* suggests the need for combination treatment with macrolides or quinolones. *M. pneumoniae* was highly detected during AE-COPD in patients who used ICS, ICS/LABA, LAMA, or no inhaler. Epidemics tend to occur in patient with higher social activity because *M. pneumoniae* is spread via respiratory droplets, epidemics frequently arise among persons living in close quarters [36,37]. It can be thought that mycoplasma infection may be higher because patients using those inhalers have relatively preserved lung function and have a higher ratio of A than COPD group D and they often maintain social activity.

In our study, the risk of RSV infection in the ICS group was lower than in the non-ICS group during AE-COPD. In several previous studies, the viral detection rate during AE-COPD was approximately 22–64% [35,38,39,40,41,42,43,44], compared to 33% in the present study. In our study, rhinovirus and influenza A virus were the most commonly detected viruses, compared to RSV and coronavirus in previous studies [35,38,39,40,41,42,43,44]. In a study of 192 patients with AE-COPD who required hospitalization [45], the virus detection rate was higher in patients with previous ICS use. However, the virus detection rate during AE-COPD was lower in severe COPD patients who used ICS compared to patients who did not use ICS, although the difference was not statistically significant. In a mouse experiment [46], nasal steroid spray increased the replication rate of respiratory viruses. In a previous study of primary culture of human tracheal epithelial cells [47] infected with rhinovirus, formoterol and budesonide inhibited rhinovirus infection by reducing intercellular adhesion molecule-1 levels and/or acidic endosomes, and modulated airway inflammation associated with rhinovirus infections. Our hypothesis is that the immunological response to bacteria and viruses is different, and bacteria usually cause a lot of indolent colonization in COPD patients, and in AE-COPD, these indolent colonization bacteria could act as a pathogen causing active infection. Further studies are required to determine the mechanism underlying the viral inhibition induced by ICS.

Our study has several limitations. The first is that accurate evaluation of past exacerbation history and symptoms could not be achieved due to a retrospective study. Drugs are used in COPD patients based on the symptoms and exacerbation history; therefore, we adjusted for the effect of past exacerbations on drug use based on the GOLD grade, as determined from the exacerbation history and lung function. Second, detailed information regarding the drug dose and administration duration could not be obtained. Recent studies have evaluated drug dose- and eosinophil count-related risks for pneumonia and other causative infections of AE-COPD [22,45,48]. Additional studies are required to determine the optimal ICS dose. Third, this was a retrospective multicentre study and there were differences in pathogen investigations among the study centres. Forth, there is no data on colonization of *P. aeruginosa* in the airway in stable status of our study, so it was not possible to distinguish between colonization and active infection *P. aeruginosa*. However, the strongest predictor for *P. aeruginosa* infection is prior isolation of this species in sputum culture [24].

In conclusion, we analysed the microbiological characteristics of AE-COPD patients according to previous drug use. The *P. aeruginosa* detection rate was highest among all pathogens during AE-COPD. Although the overall bacterial detection rate was higher in the ICS than non-ICS group, the use of ICS did not significantly increase the detection rate of viruses or drug-resistant bacteria during AE-COPD. However, the RSV detection rate was lower in the ICS than non-ICS group after adjusting for COPD severity. However, COPD patients who used systemic steroids for 6 months before AE-COPD had a higher *P. aeruginosa* infection rate than those who did not use systemic steroids. Therefore, patients with AE-COPD who have used systemic steroids within the previous 6 months should be treated with antibiotics that have activity against *P. aeruginosa*.

## Figures and Tables

**Figure 1 jcm-12-00111-f001:**
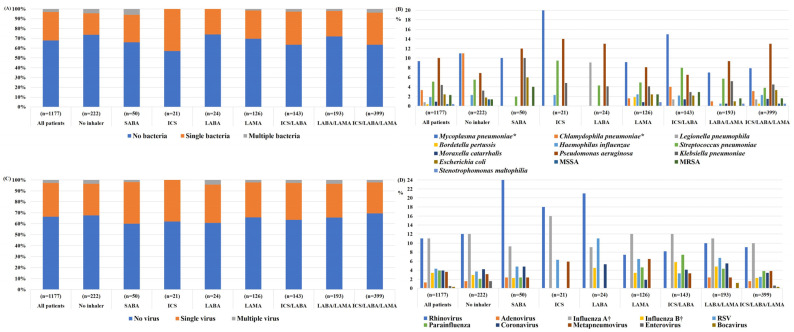
Bacterial and viral detection rates during acute exacerbations of chronic obstructive pulmonary disease according to the drug used: (**A**) overall bacterial detection rate, (**B**) detection rate for each bacterial species, (**C**) overall virus detection rate, and (**D**) detection rate for each virus species. Values are presented as percentages. SABA, short-acting beta-agonist; ICS, inhaled corticosteroid; LABA, long-acting beta 2-agonist; LAMA, long-acting muscarinic antagonist. * Bacteria were detected in respiratory specimens via polymerase chain reaction, or in serum via immunoglobulin M measurement (n = 739). † Viruses were detected in respiratory specimen using immunofluorescence assay or polymerase chain reaction (n = 1131). MSSA, methicillin-sensitive *Staphylococcus aureus*; MRSA, methicillin-resistant *Staphylococcus aureus*; RSV, respiratory syncytial virus.

**Figure 2 jcm-12-00111-f002:**
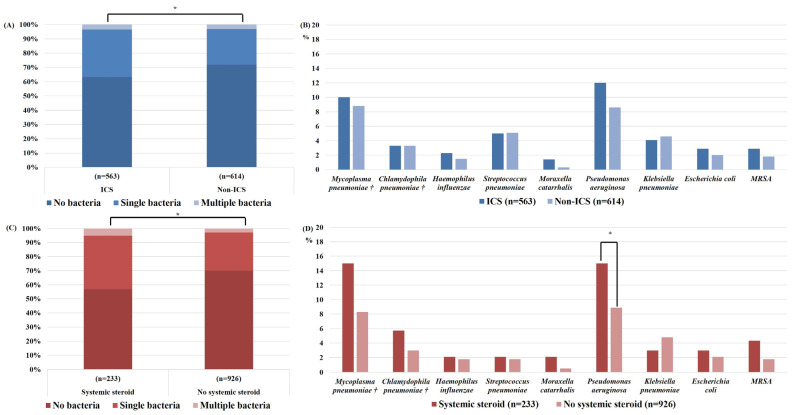
Bacterial detection rates during acute exacerbations of chronic obstructive pulmonary disease, with or without ICS or systemic steroid used within the previous 6 months: (**A**) overall bacterial detection rate, (**B**) detection rate for each bacterial species with or without ICS use, (**C**) overall bacterial detection rate, and (**D**) detection rate for each bacterial species with or without systemic steroid use. * *p* < 0.05, † Bacteria were detected in respiratory specimens via polymerase chain reaction, or in serum via immunoglobulin M measurement (n = 602). ICS, inhaled corticosteroid; MRSA, methicillin-resistant *Staphylococcus aureus*.

**Figure 3 jcm-12-00111-f003:**
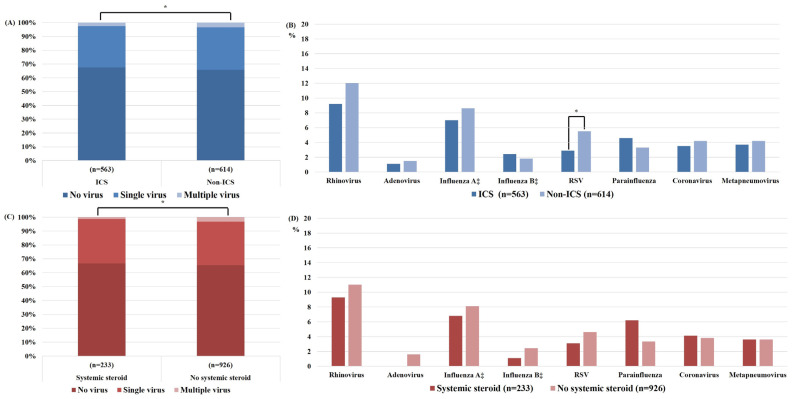
Virus detection rates during acute exacerbations of chronic obstructive pulmonary disease with or without ICS or systemic steroid use: (**A**) overall virus detection rate, (**B**) detection rate for each virus species with or without ICS use, (**C**) overall virus detection rate, and (**D**) detection rate for each virus species with or without systemic steroid use. * *p* < 0.05, ‡ Viruses were detected in respiratory specimens using immunofluorescence assay or polymerase chain reaction (n = 974). ICS, inhaled corticosteroid; RSV, respiratory syncytial virus.

**Table 1 jcm-12-00111-t001:** Baseline demographic and clinical characteristics of patients with acute exacerbation of chronic respiratory disease according to type of inhaler.

	All Patients (n = 1177)	No Inhaler (n = 222)	SABA (n = 50)	ICS (n = 21)	LABA (n = 24)	LAMA (n = 126)	ICS/LABA (n = 143)	LABA/LAMA (n = 193)	ICS/LABA/LAMA (n = 399)	*p*-Value
Sex, male	974 (83%)	188 (85%)	44 (88%)	17 (81%)	19 (83%)	102 (81%)	112 (78%)	157 (81%)	335 (84%)	0.718
Age, years	75 (69–80)	75 (69–80)	77 (71–81)	80 (73–85)	75 (62–82)	77 (72–81)	75 (67–81)	74 (68–80)	74 (68–79)	0.023
BMI, kg/m^2^	21.3 (18.8–23.9)	21.5 (19.1–24.3)	21.0 (18.7–23.7)	20.2 (17.3–21.7)	22.2 (17.5–24.9)	21.1 (18.7–24.0)	21.4 (18.7–24.0)	21.3 (18.6–23.6)	21.3 (18.8–23.9)	0.773
Disease duration	6.0 (2.5–11.0)	4.0 (1.0–8.0)	4.5 (0.1–10.0)	5.0 (2.5–10.0)	4.0 (2.0–7.3)	6.0 (2.0–10.0)	5.0 (2.0–10.0)	5.0 (2.0–10.0)	8.3 (4.0–15.0)	<0.001
Treatment duration	5.0 (2.0–10.0)	3.0 (1.0–7.0)	4.0 (0.2–10.0)	4.0 (2.5–7.0)	4.0 (2.0–7.3)	5.0 (2.0–10.0)	5.0 (2.0–10.0)	5.0 (2.0–10.0)	8.0 (4.0–14.0)	<0.001
Smoking history										0.001
Never smoker	312 (27%)	69 (32%)	12 (25%)	10 (48%)	8 (35%)	43 (35%)	39 (28%)	48 (26%)	83 (21%)	
Current smoker	153 (13%)	38 (17%)	8 (16%)	2 (10%)	1 (4.3%)	18 (15%)	23 (16%)	28 (15%)	35 (15%)	
Ex-smoker	686 (60%)	112 (51%)	29 (59%)	9 (43%)	14 (61%)	61 (50%)	80 (56%)	110 (60%)	271 (70%)	
Pack-years	40.0 (21.5–50.0)	36.5 (20.0–50.0)	30.0 (20.0–50.0)	30.0 (5.0–40.0)	30.0 (20.0–44.0)	40.0 (20.0–50.0)	40.0 (18.8–50.0)	40.0 (20.0–50.0)	40.0 (25.0–50.0)	0.081
Previous respiratory disease										
Tuberculosis	375 (32%)	69 (31%)	15 (30%)	6 (29%)	12 (52%)	44 (35%)	37 (26%)	72 (37%)	120 (30%)	0.150
Bronchiectasis	168 (14%)	35 (16%)	4 (8.0%)	1 (4.8%)	3 (13%)	13 (10%)	21 (15%)	29 (15%)	62 (16%)	0.559
Interstitial lung disease	27 (2.3%)	9 (4.1%)	1 (2.0%)	0	1 (4.3%)	2 (1.6%)	0	4 (2.1%)	10 (2.5%)	0.353
Co-morbidities										
Diabetes mellitus	317 (26%)	75 (34%)	16 (32%)	6 (29%)	3 (13%)	31 (25%)	39 (27%)	32 (25%)	115 (29%)	0.006
Hypertension	580 (49%)	110 (49%)	30 (60%)	13 (62%)	15 (65%)	62 (49%)	77 (54%)	87 (45%)	186 (47%)	0.208
Congestive heart disease	166 (14%)	36 (16%)	7 (14%)	3 (14%)	4 (17%)	19 (15%)	19 (13%)	11 (6%)	67 (17%)	0.038
Chronic kidney disease	75 (6.4%)	22 (10%)	2 (4.0%)	2 (9.5%)	6 (26%)	5 (4.0%)	10 (7.0%)	12 (6.2%)	16 (4.0%)	0.001
Cerebrovascular disease	70 (5.9%)	19 (8.6%)	5 (10%)	1 (4.8%)	1 (4.3%)	9 (7.1%)	12 (8.4%)	12 (6.2%)	11 (2.8%)	0.062
Advanced cancer	138 (12%)	23 (10%)	5 (10%)	0	1 (4.3%)	18 (14%)	18 (13%)	25 (13%)	48 (12%)	0.568
Lung function (n = 898)										
FEV1, L	1.07 (0.77–1.50)	1.20 (0.81–1.91)	1.14 (0.59–1.50)	1.34 (0.95–1.61)	1.25 (1.03–1.65)	1.18 (0.78–1.68)	1.21 (0.86–1.57)	1.17 (0.90–1.59)	0.94 (0.71–1.32)	< 0.001
FEV1, % predicted	47 (33–63)	53 (37–73)	42 (28–59)	53 (43–70)	52 (38–72)	56 (33–70)	53 (36–68)	49 (37–64)	39 (29–54)	<0.001
Bronchodilator response	166 (14%)	17 (8%)	5 (10%)	8 (40%)	4 (19%)	8 (7%)	21 (15%)	22 (11%)	81 (21%)	<0.001
GOLD group										<0.001
A	63 (5.4%)	16 (7.2%)	0	0	2 (8.7%)	11 (8.7%)	6 (4.2%)	17 (8.8%)	11 (2.8%)	
B	140 (12%)	24 (11%)	8 (16%)	3 (14%)	5 (22%)	16 (13%)	12 (8%)	33 (17%)	39 (10%)	
C	40 (3.4%)	2 (0.9%)	0	2 (10%)	0	0	10 (5.2%)	10 (5.2%)	16 (4.0%)	
D	224 (19%)	4 (1.8%)	5 (10%)	1 (4.8%)	8 (35%)	15 (12%)	31 (22%)	40 (21%)	120 (30%)	
Leukotriene receptor antagonist	216 (18%)	12 (5.4%)	2 (4.1%)	4 (19%)	4 (17%)	12 (10%)	31 (22%)	29 (15%)	122 (31%)	<0.001
Xanthine derivative	365 (31%)	31 (14%)	8 (16%)	9 (43%)	5 (22%)	40 (32%)	45 (32%)	55 (29%)	172 (43%)	<0.001
Systemic steroid	238 (20%)	23 (10%)	11 (22%)	9 (43%)	5 (22%)	19 (15%)	38 (27%)	30 (16%)	103 (26%)	<0.001

Values are presented as number (%) or median value (interquartile range). SABA: short-acting beta 2-agonist; ICS, inhaled corticosteroid; LABA, long-acting beta 2-agonist; LAMA, long-acting muscarinic antagonist; BMI, body mass index; FEV_1_, forced expiratory volume in one second; GOLD, Global Initiative for Obstructive Lung Disease.

**Table 2 jcm-12-00111-t002:** Logistic regression analysis of the effect of inhaled corticosteroid or systemic steroid administration on bacterial or viral detection during exacerbation of chronic obstructive pulmonary disease adjusted for the severity of COPD.

	Inhaled Corticosteroid			Systemic Steroid		
	OR	CI	*p*-Value	OR	CI	*p*-Value
Bacteria						
*Mycoplasma pneumoniae* ^†^	1.134	0.624–2.061	0.680	1.876	0.952–3.700	0.069
*Chlamydophila pneumoniae* ^†^	1.154	0.407–3.277	0.787	2.635	0.773–8.986	0.122
*Haemophilus influenzae*	1.395	0.563–3.458	0.472	1.097	0.387–3.105	0.962
*Streptococcus pneumoniae*	0.890	0.502–1.577	0.689	1.728	0.922–3.240	0.088
*Moraxella catarrhalis*	2.873	0.582–14.188	0.195	2.444	0.629–9.505	0.197
*Pseudomonas aeruginosa*	1.238	0.796–1.927	0.343	1.619	1.007–2.603	0.047
*Klebsiella pneumoniae*	0.843	0.459–1.550	0.583	0.698	0.303–1.610	0.399
*Escherichia coli*	1.792	0.747–4.302	0.192	1.420	0.567–3.552	0.454
MRSA	1.983	0.725–5.426	0.182	0.825	0.263–2.585	0.741
Virus						
Rhinovirus	0.652	0.413–1.027	0.065	0.725	0.400–1.312	0.288
Adenovirus	0.616	0.191–1.985	0.417			
Influenza A ^‡^	0.855	0.483–1.513	0.591	0.940	0.463–1.907	0.864
Influenza B ^‡^	1.197	0.400–3.579	0.748	0.625	0.135–2.911	0.511
RSV	0.492	0.244–0.988	0.045	0.670	0.271–1.655	0.386
Parainfluenza	1.068	0.528–2.159	0.854	1.848	0.865–3.948	0.113
Coronavirus	0.942	0.452–1.963	0.873	0.906	0.359–2.287	0.835

^†^ Bacteria were detected in respiratory specimens via polymerase chain reaction or in serum via immunoglobulin M measurement. (n = 602); ^‡^ Virus was detected in respiratory specimen using immunofluorescence assay or polymerase chain reaction. (n = 974); OR, odds ratio; CI, confidence interval; MRSA, methicillin-resistant Staphylococcus aureus; RSV, respiratory syncytial virus.

## Data Availability

The data presented in this study are available on request from the corresponding author.

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
