# Peer review of "COPD Exacerbation-Related Pathogens and Previous COPD Treatment"

_jcm, 2022, doi:10.3390/jcm12010111_

Round 1

Reviewer 1 Report

Major comments:

In the present report, the authors conducted a retrospective study to investigate whether the microbiological pathogens identified during AE-COPD are associated with the COPD medications used before AE-COPD. They demonstrated that COPD patients who used systemic steroids within previous 6 months before AE-COPD had a higher P. aeruginosa infection rate than those who did not, indicating that patients with AE-COPD who have used systemic steroids within previous 6 months would be treated with antibiotics that have activity against P. aeruginosa. These observations are potentially interesting, and the manuscript is moderately well written. To enhance the priority, following several points require attention and revision.

1)      The bacterial results in patients with AE-COPD according to the type of inhaler were shown in Figure 1. P. aeruginosa had the highest detection rates in the ICS/LABA/LAMA, LABA/LAMA, LABA, and SABA groups, whereas M. pneumoniae had the highest detection rates in the ICS/LABA, LAMA, ICS, and no inhaler use groups. What were the reasons for these differences of bacterial detection among COPD medications? The authors would be recommended to add some comments.

2)      The P. aeruginosa detection rate was increased with systemic steroid use and the rate of RSV detection was decreased with ICS use after adjusting for COPD severity. However, there were no changes of detection rates among the type of previous COPD medications in other pathogens. The authors would be recommended to discuss the putative cause of these discrepancies between P. aeruginosa, RSV and other microorganisms.

3)      In the present study, the P. aeruginosa detection risk was significantly increased in patients using systemic steroid after adjusting for COPD severity. However, the detection of P. aeruginosa is not necessarily considered as active infection, that is a causative pathogen, because P. aeruginosa is one of the main microbial species colonizing the lungs of various chronic pulmonary diseases as well known. Did the authors differentiate between active infection and simple colonization of P. aeruginosa?

4)      Were there any differences of prognoses of enrolled COPD patients among microorganisms identified during AE-COPD? Was the prognosis of AE-COPD patients with P. aeruginosa infection worse than the others?

5)      How the authors define “previous COPD medications” as medications used “within 6 months before AE-COPD”?

Author Response

Reviewer’s comments

Manuscript ID: jcm-2076996

Author: Yun Su Sim et al.

We sincerely appreciate the time and effort you invested to help us improve this manuscript. We revised the manuscript (red color) and have provided a point-by-point response to your comments below.

Major comments:

In the present report, the authors conducted a retrospective study to investigate whether the microbiological pathogens identified during AE-COPD are associated with the COPD medications used before AE-COPD. They demonstrated that COPD patients who used systemic steroids within previous 6 months before AE-COPD had a higher P. aeruginosa infection rate than those who did not, indicating that patients with AE-COPD who have used systemic steroids within previous 6 months would be treated with antibiotics that have activity against P. aeruginosa. These observations are potentially interesting, and the manuscript is moderately well written. To enhance the priority, following several points require attention and revision.

  • The bacterial results in patients with AE-COPD according to the type of inhaler were shown in Figure 1. aeruginosa had the highest detection rates in the ICS/LABA/LAMA, LABA/LAMA, LABA, and SABA groups, whereas M. pneumoniae had the highest detection rates in the ICS/LABA, LAMA, ICS, and no inhaler use groups. What were the reasons for these differences of bacterial detection among COPD medications? The authors would be recommended to add some comments.

A1. Thanks for pointing this important issue. We add “M. pneumoniae was highly detected during AE-COPD in patients who used ICS, ICS/LABA, LAMA, or no inhaler. Epidemics tend to occur in patient with higher social activity because M. pneumoniae is spread via respiratory droplets, epidemics frequently arise among persons living in close quarters. It can be thought that mycoplasma infection may be higher because patients using those inhalers have relatively preserved lung function and have a higher ratio of A than COPD group D and they often maintain social activity” in discussion

New reference

  1. Foy, H.M.; Kenny, G.E.; Cooney, M.K.; Allan, I.D. Long-term epidemiology of infections with Mycoplasma pneumoniae. J Infect Dis 1979, 139, 681-687, doi:10.1093/infdis/139.6.681.
  2. Walter, N.D.; Grant, G.B.; Bandy, U.; Alexander, N.E.; Winchell, J.M.; Jordan, H.T.; Sejvar, J.J.; Hicks, L.A.; Gifford, D.R.; Alexander, N.T., et al. Community outbreak of Mycoplasma pneumoniae infection: school-based cluster of neurologic disease associated with household transmission of respiratory illness. J Infect Dis 2008, 198, 1365-1374

  • The aeruginosa detection rate was increased with systemic steroid use and the rate of RSV detection was decreased with ICS use after adjusting for COPD severity. However, there were no changes of detection rates among the type of previous COPD medications in other pathogens. The authors would be recommended to discuss the putative cause of these discrepancies between P. aeruginosa, RSV and other microorganisms.

A2. Thanks for good advises. Long-term treatment with corticosteroids weakens adaptive immune response by down-modulating MHC class II and costimulatory molecules. Systemic steroid use was also reported to be associated with poor clearance of causative microbes of acute exacerbation following antibiotic treatment. There are few references on the difference in the detection of bacteria and viruses according to the use of systemic steroids and ICS in AE-COPD, so it was very difficult to interpret this part. Our hypothesis is that the immunological response to bacteria and viruses is different, and bacteria usually cause a lot of indolent colonization in COPD patients, and in AE-COPD, these indolent colonization bacteria could act as a pathogen causing active infection. In a previous study of primary culture of human tracheal epithelial cells infected with rhinovirus, formoterol and budesonide inhibited rhinovirus infection by reducing intercellular adhesion molecule-1 levels and/or acidic endosomes, and modulated airway inflammation associated with rhinovirus infections. We revise these contents by adding " Long-term use with steroids weakens adaptive immune response by down-regulation MHC class II and costimulatory molecules. Systemic steroid treatment was also reported to be associated with poor clearance of causative microbiologic pathogen of AE-COPD following antibiotic treatment. Our hypothesis is that the immunological response to bacteria and viruses is different, and bacteria usually cause a lot of indolent colonization in COPD patients, and in AE-COPD, these indolent colonization bacteria could act as a pathogen causing active infection. " to the discussion section.

New reference

  1. Sethi, S.; Anzueto, A.; Miravitlles, M.; Arvis, P.; Alder, J.; Haverstock, D.; Trajanovic, M.; Wilson, R. Determinants of bacteriological outcomes in exacerbations of chronic obstructive pulmonary disease. Infection 2016, 44, 65-76, doi:10.1007/s15010-015-0833-3.
  • In the present study, the aeruginosa detection risk was significantly increased in patients using systemic steroid after adjusting for COPD severity. However, the detection of P. aeruginosais not necessarily considered as active infection, that is a causative pathogen, because P. aeruginosa is one of the main microbial species colonizing the lungs of various chronic pulmonary diseases as well known. Did the authors differentiate between active infection and simple colonization of P. aeruginosa?

A2. Thanks for good advises. We fully agree with your opinion. Unfortunately, there is no data on colonization of P. aeruginosa in stable status of COPD patients of our study, so it was not possible to distinguish between colonization and active infection P. aeruginosa. However, a study for the bronchial microbiome in severe COPD during stability and exacerbation in patients chronically colonized by P.aeruginosa reported that stable COPD patients with severe disease and P.aeruginosa colonized showed a similar biodiversity to non- P.aeruginosa-colonized patients, with a higher relative abundance of Pseudomonas genus in bronchial secretions. Exacerbation in severe COPD patients showed the same microbial pattern, independently of previous colonization by P. aeruginosa. In addition, although P.pneumonia detected in AE-COPD cannot accurately distinguish between colonization and acute infection, it may be clinical usefulness to consider choice of antibacterial treatment. In addition, the strongest predictor for P. aeruginosa infection is prior isolation of this species in sputum culture. The contents of the previous pseudomonas colonization of your review and advise were added to the limitations in the discussion section. “Forth, there is no data on colonization of P. aeruginosa in the airway in stable status of our study, so it was not possible to distinguish between colonization and active infection P. aeruginosa. However, the strongest predictor for P. aeruginosa infection is prior isolation of this species in sputum culture.

New reference

  1. Millares, L.; Ferrari, R.; Gallego, M.; Garcia-Nunez, M.; Perez-Brocal, V.; Espasa, M.; Pomares, X.; Monton, C.; Moya, A.; Monso, E. Bronchial microbiome of severe COPD patients colonised by Pseudomonas aeruginosa. Eur J Clin Microbiol Infect Dis 2014, 33, 1101-1111, doi:10.1007/s10096-013-2044-0.

  • Were there any differences of prognoses of enrolled COPD patients among microorganisms identified during AE-COPD? Was the prognosis of AE-COPD patients with aeruginosa infection worse than the others?

A5. Thanks for pointing this important issue. Unfortunately, our study is a retrospective study and has limitations in analyzing prognosis of AE-COPD patients with microbiological results including P. aeruginosa. However, it would be interesting to analyze the relationship between prognosis and microbiological results of AE-COPD patients in future studies.

  • How the authors define “previous COPD medications” as medications used “within 6 months before AE-COPD”?

A5. Thanks for pointing this important issue. I agree that “previous” of the sentence may confuse the reader. Therefore, "medicine used for 6 months before AE-COPD." added in the introduction and methods section for define a clear period for "previous".

Reviewer 2 Report

I evaluated the article entitled "COPD exacerbation-related pathogens and previous COPD treatment" submitted for publication in the Journal of Clinical Medicine. Although the study has emerged with an important clinical question, it has serious methodological problems. Most of the statistical analyzes made and expressed are inaccurate. I think that the data can only be expressed as a simple descriptive study by eliminating all hypothesis tests. My suggestions are stated below:

- Introduction: In the Introduction paragraphs, important information about the treatment of COPD is given and the role of infectious pathogens in the course of the disease is mentioned. In the last paragraph, these pathogens are tried to be associated with inhaled corticosteroids use. This part is not understood. The hypothesis tried to be established in the last paragraph should be explained more clearly.

- Method: The type of study should be specified in the Design section. The sections related to the missing data in the Methods should be transferred to the Results section.

- Method: What is the primary endpoint of the study?

- Results: Which comparison does the p values ​​indicated in Table 1 belong to? It is observed that some of the values ​​given as mean (standard deviation) in the same table (for example, pack-year) do not conform to the normal distribution. For these, the appropriate analysis method should be selected and expressed with the median (IQR).

- Results: Frequent inhaler changes can be made in COPD patients. For this reason, it is difficult to observe such a clear relationship between inhaler groups and a cross-sectional microbiological examination of patients. However, in the statistical analysis performed with such a large number of subgroups, many posthoc tests are required to understand which group the difference originates from. It is not possible to reach these results with the sample size in question.

Regards.

Author Response

Reviewer 2

We sincerely appreciate the time and effort you invested to help us improve this manuscript. We revised the manuscript (red color) and have provided a point-by-point response to your comments below.

I evaluated the article entitled "COPD exacerbation-related pathogens and previous COPD treatment" submitted for publication in the Journal of Clinical Medicine. Although the study has emerged with an important clinical question, it has serious methodological problems. Most of the statistical analyzes made and expressed are inaccurate. I think that the data can only be expressed as a simple descriptive study by eliminating all hypothesis tests. My suggestions are stated below:

- Introduction: In the Introduction paragraphs, important information about the treatment of COPD is given and the role of infectious pathogens in the course of the disease is mentioned. In the last paragraph, these pathogens are tried to be associated with inhaled corticosteroids use. This part is not understood. The hypothesis tried to be established in the last paragraph should be explained more clearly.

A1. Thanks for your advice. We fully agree with your opinion. We revise by repositioning " The treatment of stable COPD is based on the inhalation of long-acting bronchodilators alone or in combination with long-acting beta 2-agonist (LABA), and long-acting muscarinic antagonist (LAMA). Inhaled corticosteroid (ICS) with anti-inflammatory action is used in combination with other inhaled bronchodilators to improve lung function and health status and reduce exacerbations in patients with moderate to very severe COPD." and the importance of infectious etiology in AE-COPD in the Introduction. And there are various factors that affect infectious etiology in AE-COPD, and one of them suggested the possibility of medication usual taken.

“Since the infectious pathogen of COPD patients can be affected by the duration of the disease, the patient's comorbidties, and the previous exacerbation history, various factors of AE- COPD patients should be considered when selecting antibiotics for appropriate treatment. Previous medications used in COPD patients could also be important factors affecting respiratory pathogens detected in AE-COPD”.

After that, after mentioning reports on respiratory pathogens and commonly used drugs including ICS found in AE-COPD in other studies, the goal in our study is to investigate the relationship between commonly used drugs in COPD patients and infectious etiology detected in AE-COPD.

- Method: The type of study should be specified in the Design section. The sections related to the missing data in the Methods should be transferred to the Results section.

A2: Thank you for your advice. We added “This study is retrospective multicenter cohort study” in methods section. We revise the sections related to the missing data (Mycoplasma pneumoniae and Chlamydophila pneumoniae were detected in respiratory specimens via PCR or in serum via immunoglobulin M analysis. The tests were performed in 739 cases, and cases that were not tested were treated as missing data. Influenza A and B were detected in respiratory specimen using immunofluorescence assay or PCR. The above tests were performed in 1,131 cases were tested, and cases that were not tested were treated as missing data.) in the methods by transfer them to results.

New reference

  1. Park, Y.B.; Rhee, C.K.; Yoon, H.K.; Oh, Y.M.; Lim, S.Y.; Lee, J.H.; Yoo, K.H.; Ahn, J.H.; Committee of the Korean, C.G. Revised (2018) COPD Clinical Practice Guideline of the Korean Academy of Tuberculosis and Respiratory Disease: A Summary. Tuberc Respir Dis (Seoul) 2018, 81, 261-273, doi:10.4046/trd.2018.0029.
  2. Lee, H.W.; Sim, Y.S.; Jung, J.Y.; Seo, H.; Park, J.W.; Min, K.H.; Lee, J.H.; Kim, B.K.; Lee, M.G.; Oh, Y.M., et al. A Multicenter Study to Identify the Respiratory Pathogens Associated with Exacerbation of Chronic Obstructive Pulmonary Disease in Korea. Tuberc Respir Dis (Seoul) 2022, 85, 37-46, doi:10.4046/trd.2021.0080.
  3. Seo, H.; Sim, Y.S.; Min, K.H.; Lee, J.H.; Kim, B.K.; Oh, Y.M.; Ra, S.W.; Kim, T.H.; Hwang, Y.I.; Park, J.W. The Relationship Between Comorbidities and Microbiologic Findings in Patients with Acute Exacerbation of Chronic Obstructive Pulmonary Disease. Int J Chron Obstruct Pulmon Dis 2022, 17, 855-867, doi:10.2147/COPD.S360222.

- Method: What is the primary endpoint of the study?

A3. Thank you for your advice. We investigated whether the pathogens identified during AE-COPD are associated with the COPD medications previous used. It is difficult to clearly define the primary endpoint because our study could not be analyzed in terms of effects, but what we want to evaluate is the difference in microbiologic results in AE-COPD according to the previous drug used. If the microbiologic result of AE-COPD according to these drugs can be known, it will be helpful for the choice of antibacterial or antiviral treatment according to the patient's past use of drugs.

- Results: Which comparison does the p values ​​indicated in Table 1 belong to? It is observed that some of the values ​​given as mean (standard deviation) in the same table (for example, pack-year) do not conform to the normal distribution. For these, the appropriate analysis method should be selected and expressed with the median (IQR).

A4. Thank you for your advice. In Table 1, we revise continuous variables as comparisons by non-parametric tests and median values (interquartile range). In the statics of the methods section, it was revised that continuous variables were analyzed with the Kruskal-wallis test.

- Results: Frequent inhaler changes can be made in COPD patients. For this reason, it is difficult to observe such a clear relationship between inhaler groups and a cross-sectional microbiological examination of patients. However, in the statistical analysis performed with such a large number of subgroups, many posthoc tests are required to understand which group the difference originates from. It is not possible to reach these results with the sample size in question.

A5. Thank you for your comment. We fully agree with your opinion. When we first started the study, we wanted to show statistically significant differences in microbiologic results for each drug, but it was difficult to obtain statistically significant results because there were many types of drugs used and many types of bacteria and viruses cultured in each. However, we believe that showing the tendency of highly cultured bacteria and viruses for each drug can be a clinically valuable result. In addition, in order to compensate for the fact that the sample size is not statistically significant due to the large number of drugs, the entire data was analyzed by grouping into groups using systemic steroids or ICS and groups not using them. Corticosteroid is a drug that can most affect the microbiological results of AE-COPD patients because corticosteroids are immunologically diverse drugs, and only comparison between the two groups could statistically compensate for the sample size.
